# Comparison of Ultrasound and Magnetic Resonance Imaging for Identifying Soft Tissue Abnormalities in the Palmar Aspect of the Equine Digit

**DOI:** 10.3390/ani13142328

**Published:** 2023-07-17

**Authors:** Myra F. Barrett, Georgette E. Goorchenko, David D. Frisbie

**Affiliations:** 1Department of Environmental and Radiological Health Sciences, College of Veterinary Medicine and Biomedical Sciences, Colorado State University, Fort Collins, CO 80523, USA; 2Great & Small Veterinary Imaging, Inc., 1437 Moreno St, Oceanside, CA 92054, USA; georgette@greatandsmallimaging.com; 3Department of Clinical Sciences, College of Veterinary Medicine and Biomedical Sciences, Colorado State University, Fort Collins, CO 80523, USA; dfrisbie@colostate.edu

**Keywords:** deep digital flexor tendon, navicular bone, navicular bursa, horse, navicular apparatus, MRI, podotrochlear apparatus, equine diagnostic imaging

## Abstract

**Simple Summary:**

Lameness originating from the distal thoracic limb is common in horses and is often due to damage to the soft tissue structures in the heel region. The hoof capsule limits access to this region with ultrasound, and the scan approach can be technically challenging. However, because ultrasound is more readily available and less expensive than MRI, it is often used to assess injuries in this area. We compared ultrasound and high-field MRI findings of the palmar digit of 45 horses. MRI was considered the gold standard for the lesions. We found that ultrasound was a useful screening tool for assessing damage to the deep digital flexor tendon but risked overdiagnosing injuries. It also risks underdiagnosing injuries to other soft tissues in the palmar digit region.

**Abstract:**

Damage to the soft tissue structures of the digit is a common source of equine lameness. While magnetic resonance imaging (MRI) allows for the most complete diagnostic imaging of the equine digit, ultrasound is more readily available and less expensive. This prospective diagnostic accuracy study compares ultrasound to MRI for the diagnosis of injuries visible with ultrasound within the digit, including the deep digital flexor tendon (DDFT), collateral sesamoidean ligament (CSL), and navicular bursa. Clinical patients underwent an MRI of the digit and a blinded ultrasound of the digit between the heel bulbs, and results of the two modalities were compared. A total of 70 ultrasound and MRI exams of 45 horses were included. Ultrasound had good sensitivity (85%), moderate specificity (60%), and accuracy of 70% for evaluating the dorsal tearing of the DDFT. Accuracy was lower for navicular bursa effusion (67%), navicular bursa proliferation (61%), and CSL enlargement (61%). Tearing of the DDFT distal to the navicular bone was identified with MRI in 27 limbs, 20 of which also had dorsal damage proximal to the navicular bone identified with ultrasound. Ultrasound evaluation remains a useful screening tool, particularly for the assessment of DDFT tearing proximal to the navicular bone but risks under-diagnosing pathology to the navicular bursa and CSL. Clinically significant concurrent damage to the distal DDFT and other osseous and soft tissues in the hoof capsule is unlikely to be identified without MRI.

## 1. Introduction

Damage of soft tissue and osseous structures within the digit is a common cause of lameness in the equine athlete. Although magnetic resonance imaging (MRI) has greatly improved diagnostic capabilities for imaging the digit, its use remains limited for multiple reasons, including cost and access. Ultrasound is more available, less costly, and does not require general anesthesia. However, even with a skilled sonographer, diagnostic ultrasound of the digit has limitations due to the depth and orientation of the anatomic structures and varies between horses due to digit conformation. In addition, many of the structures of the digit cannot be evaluated with ultrasound due to their location within the hoof capsule.

The normal and abnormal appearance of the structures within the digit on ultrasound [1,2,3,4,5,6,7,8] and MRI [9,10,11,12,13,14,15,16,17,18,19,20,21,22,23,24,25,26,27] has been described, including ultrasonographic evaluation through the frog [28,29]. Dyson et al. [22], evaluated lesions located within the hoof capsule and their appearance and association with navicular pathology diagnosed by MRI. Lesions of the deep digital flexor tendon were less common at the level of the pastern joint and more common distally within the hoof capsule. Navicular bone pathology was also associated with abnormalities of multiple other structures within the digit, including the deep digital flexor tendon (DDFT), collateral sesamoidean ligament (CSL), impar ligament, and pathology of the distal interphalangeal (DIP) joint. Distal injuries to the DDFT were shown to have clinical significance in another study by Dyson et al. [21]. Horses with an injury to the DDFT in combination with the navicular bone had a much worse prognosis for return to athletic function. This raises the question of whether a positive ultrasound finding of injury to the DDFT at the level of the proximal recess of the navicular bursa underestimates pathology distally in the digit. Information about the extent of injury/pathology may alter the prognosis and helps to manage the expectations of the owner and treating veterinarian.

Due to the clinical importance of accurately diagnosing injuries within the digit and the use of both ultrasound and MRI for this purpose, it is important to understand the value and limitations of ultrasound compared to MRI for the evaluation of lesions within the digit. Our hypothesis is that ultrasound will underestimate the presence and extent of injury to the soft tissue structures within the digit. The purpose of this study is to compare ultrasound to MRI for the detection of abnormalities within the palmar tissues of the digit, including the DDFT at the level of the proximal recess of the navicular bursa, the CSL, and the proximal recess of the navicular bursa. In addition, our purpose is to describe the incidence of additional findings contained within the hoof capsule that was not imaged with ultrasound.

## 2. Materials and Methods

Horses were included that had MRI and ultrasound examinations of the digit between October 2014 to October 2016. Horses were included if a forelimb lameness was localized to the distal extremity by palmar digital, basisesamoid, or abaxial perineural anesthesia. Ultrasound exams were performed by two of the authors experienced in equine ultrasound (one boarded radiologist and one diagnostic imaging resident). Ultrasound exams were performed either before the MRI or following MRI but without knowledge of the MRI findings. Ultrasound evaluations were performed within 36 h of the MRI. Repeat ultrasound examination post-MRI was performed in selected cases to further characterize the visibility of certain lesions with ultrasound.

Ultrasound evaluation was performed using a GE Logiq 9 and a GE NextGen (GE Ultrasound, Milwaukee, WI 53201, USA) ultrasound machine, and an evaluation of the soft tissues of the pastern and structures of the DIP joint using a multifrequency linear transducer and the palmar distal tissue of the pastern between the heel bulbs using a microconvex transducer was included. The study was performed in a very arid climate, which markedly limits transcuneal evaluation of the structures distal to the navicular bone without significant patient preparation; therefore, this was not included in this study. Ultrasound exams were graded for enlargement, tearing, or change in echogenicity of the DDFT or CSL. The navicular bursa was graded for effusion and proliferation as none, mild, moderate, or severe (Grade 0–3) (Table 1). The collateral ligaments of the distal interphalangeal joint were assessed for enlargement and changes in echogenicity.

MRI was performed after induction of general anesthesia with the horse in lateral recumbency using a 1.0T ONI extremity magnet (GE Signa HDxT, Milwaukee, WI 53201, USA). Sequences included sagittal proton dense (PD), STIR, T1 3D gradient echo, dorsal T1 spin echo, transverse PD, STIR, oblique, and dorsal T2 spin echo, and they also included the toe to mid-pastern region. MRI exams were graded by consensus by two of the authors on a none, mild, moderate, and severe scale (Grade 0–3) (Table 1). The medial and lateral halves, or lobes, of the DDFT proximal to the navicular bone were evaluated for symmetry in size, shape, and signal intensity. Dorsal surface tearing was characterized by a discrete flap of damaged tissue. Dorsal surface fraying was defined as fiber disruption of the dorsal surface without discrete tearing. Other evaluations included assessing the severity and presence of sagittal splits or core lesions of the DDFT at the level of the proximal recess of the navicular bursa and tearing of the DDFT distal to the navicular bone. Enlargement and dorsal or palmar displacement of the CSL and soft tissue proliferation and effusion of the navicular bursa were also assessed. Additional structures evaluated with MRI included the distal interphalangeal joint, navicular bone, and impar ligament.

Statistical analysis was performed by an author trained in advanced statistics. MRI was considered the gold standard to which the ultrasound findings were compared. Sensitivity, specificity, and accuracy comparisons of ultrasound versus MRI for evaluation of the DDFT, CSL, and navicular bursa were conducted. Sensitivity was calculated by the total number of positive MRIs that also have a positive ultrasound out of the total number of positive MRs. Specificity was calculated by the total number of negative MRIs that also have a negative ultrasound out of the total number of negative ultrasounds. Accuracy was calculated by the combination of the total number of cases in which the positive MRI matched the positive ultrasound and the negative MRI matched the negative ultrasound out of the total number of cases altogether. Statistical analysis was performed using commercial software (SAS, SAS Institute Inc., Cary, NC, USA, v9.4). The proc frequency test was used for all comparisons.

## 3. Results

A total of 75 horses were included with a total of 70 MRI and ultrasound exams. A single limb was imaged in 20 horses, and both digits were imaged in 25 horses. A total of 32 right fore distal limbs and 38 left fore distal limbs were included. The mean age was 10.7 years, ranging from 5–17 years. Horses of varying uses and breeds were represented. There were 4 Thoroughbreds, 23 Quarter Horses, 15 Warmbloods, 2 Paint Horses, and 1 Arabian. Lameness information was available in 46 horses, and the average grade was 2 of 5 (using a 0–5 AAEP Lameness scale).

### 3.1. MRI and Ultrasound Comparison

Results of sensitivity, specificity, and accuracy for comparisons of the findings of the DDFT, CSL, and navicular bursa are listed in Table 2. The low number of distal interphalangeal joint collateral ligament abnormalities resulted in low power for the statistic calculation, but prevalent findings are reported below. Ultrasound had the highest sensitivity for detecting enlargement and fiber damage of the dorsal aspect of the DDFT compared to identification of dorsal tearing of the DDFT on MRI, with a sensitivity of 85%, but it was less specific (60%) (Figure 1). Ultrasound was the least sensitive versus MRI to detect displacement (sensitivity 35%) of the CSL but was most specific for this finding (81%). Ultrasound also had a low sensitivity for detecting enlargement of the CSL (42%) but a specificity of 78% (Figure 2). Similarly, ultrasound was also less sensitive to detecting navicular bursa proliferation with a sensitivity of 47% but had higher specificity (78%) (Figure 3).

As the severity level of injury to the DDFT increased, so did the sensitivity. Mild dorsal surface damage had a sensitivity of 36% (9/24), moderate damage 74% (19/24), and severe changes had 100% sensitivity (6/6).

### 3.2. Additional MRI Findings

Additional MRI findings in the palmar aspect of the digit included injury to the DDFT distal to the navicular bone, degenerative changes of the navicular bone, impar ligament desmopathy, and enthesopathy of the flexor surface of the distal phalanx. These findings were not identified with the ultrasound examinations. Degenerative injury to the navicular bone was characterized by an increased signal on STIR images (also known as bone marrow lesion or bone edema-like syndrome), erosion of the flexor cortex, enlargement of the synovial invaginations, and distal border fragments and occurred in 39 distal limb digits. The navicular bone was the most severely affected structure, graded as severe (Grade 3) in a total of seven digits of distal limbs.

A total of 27 digits had tearing of the DDFT distal to the navicular bone. Twenty of these digits (74%) had dorsal lesions on ultrasound at the level of the proximal navicular bursa recess and damage to the DDFT distal to the navicular bone. Seven digits with no evidence of tearing on ultrasound at the level of the proximal recess of the navicular bone had damage to the DDFT distal to the navicular bone on MRI.

Enthesopathy of the DDFT on the distal phalanx was identified in 15 digits, graded as mild (13), moderate (2), and severe (1). Thickening of the impar ligament was identified in 36 digits, graded as mild (18), moderate (15), and severe (3). Cartilage damage or thinning in the DIP joint was found in 19 digits, which were graded as mild in 6, moderate in 10, and severe in 3 digits.

Distal interphalangeal joint collateral ligament changes were identified on eight limbs with MRI, four of which were located distal in the hoof capsule, outside of the area of ultrasonographic visualization. Ultrasound diagnosed enlargement of four ligaments and fiber changes in nine ligaments that were considered normal on MRI. A total of eight collateral ligaments were considered abnormal on MRI; of those eight, three were also considered abnormal on ultrasound.

## 4. Discussion

Based on these results, we can accept our hypothesis that ultrasound will underestimate the degree and extent of pathology in the digit for some structures. In particular, this study highlights that even when ultrasound accurately identifies some soft tissue injuries, there can be additional soft tissue or osseous abnormalities that are not detected. In our study, twenty digits (29%) in which an abnormality was identified at the level of the proximal recess of the navicular bursa with ultrasound also had more tearing/injury that occurred distal to the navicular bone. These injuries were only identified with MRI. Additionally, a total of seven digits with no ultraonographic evidence of bulging or abnormality in the DDFT proximal to the navicular bone had tearing distally. Four of the seven were considered moderate distal tears, which would be highly likely to be a clinically significant findings. In addition to the extent of DDFT pathology that was not identified with ultrasound, there were also multiple horses with navicular bone changes that were not visible ultrasonographically. A total of 40 digits had abnormalities of the navicular bone including abnormal signal on STIR sequences or degenerative changes, of which 21 were graded as moderate or severe. While mild to moderate pathology to the navicular bone may not always be associated with tearing to the DDFT, all seven digits with severe pathology of the navicular bone on MRI also had damage to the dorsal aspect of the DDFT identified with ultrasound and MRI. Multiple other injuries were identified with MRI including cartilage damage of the DIP joint in 19 digits and desmopathy of the impar ligament in 36 digits.

We can reject the other part of our hypothesis that ultrasound will under-diagnose the amount of injury to the DDFT. While ultrasound was good for evaluating the dorsal surface tearing of the DDFT at the level of the navicular bursa, with a sensitivity of 85%, it had a lower specificity of 60%. This lower specificity indicates that we could have less ability to truly rule in a lesion of the DDFT as diagnosed by ultrasound. Of the 43/70 limbs that did not have evidence of DDFT dorsal surface tearing on MRI, 17 were over-diagnosed as having tearing or bulging of the dorsal surface on ultrasound (Figure 4). This was a surprising finding, in part because the authors tend to be conservative in their assessment of the DDFT in this location. However, imaging the navicular bursa and DDFT within the hoof capsule is limited by probe position and angle of incidence. On and off-angle imaging cannot be used to help distinguish the borders of objects located distally within the hoof capsule, and drop-out artifacts can artifactually enhance the appearance of DDFT lobe enlargement. Additionally, although the researchers are trained to be as objective as possible, there is likely some degree of unconscious bias towards suspecting abnormalities when performing an ultrasound on a horse diagnosed with pain originating from the digit.

Ultrasound also had fair to good accuracy (72%) for detecting asymmetry of the lobes of the DDFT, which can be used as an indicator for tendinopathy. In the authors’ experience, some mild degree of asymmetry of the DDFT medial and lateral lobes can occur incidentally in the pastern but is rarely found in the digit in normal horses; asymmetry of the lobes at the proximal recess of the navicular bursa is generally found in conjunction with other damage to the tendon. Ultrasound was less accurate (64%) for detecting fraying along the dorsal surface of the deep digital flexor tendon. This is less surprising as this is often a more subtle finding than discrete tearing and is further hampered by the limitations of the probe angle to assess the DDFT in this area. Further work would be beneficial to evaluate the severity of damage on MRI relative to the accuracy of detection with ultrasound.

Ultrasound was the least sensitive for detecting enlargement (42%) and displacement (35%) of the CSL, but with better specificity 78% and 81%, respectively. This was an unexpected outcome as the authors had anticipated that ultrasound would be more sensitive for evaluating the CSL. Enlargement of the CSL often presents as diffuse thickening without discrete tearing or signal change on MRI. While enlargement of the CSL may not be a primary cause of lameness, it is an important finding to recognize as it often occurs in conjunction with other injuries of the navicular apparatus. Identification of a thickened CSL could help the interpreter of the imaging to feel more confident that there could be other changes to the navicular apparatus, even if they are difficult to detect ultrasonographically. The lack of sensitivity on ultrasound is likely due in part to mild enlargement being more readily detected with MRI than ultrasound, and further investigation into the relationship between detection and severity of the thickening is warranted. Displacement of the CSL can be a useful finding to help assess surrounding changes in the synovial structures. Moderate to marked increased fluid in the navicular bursa can result in dorsal displacement of the CSL. Proliferative tissue can also cause displacement of the CSL, although this is less easily identifiable as the increased tissue in the bursa can obscure the interface between the CSL and bursa. Moderate or marked effusion of the palmar recess of the distal interphalangeal joint can cause palmar displacement of the CSL. This is a less common finding but can be helpful, particularly when trying to distinguish the origin of the palmar fluid. The authors have found that in some cases, it can be difficult to distinguish fluid in the palmar recess of the distal interphalangeal joint from fluid in the navicular bursa. If there is a greater quantity of fluid between the palmar cortex of P2 and CSL, it is within the palmar recess of the distal interphalangeal joint and can cause palmar displacement of the CSL. However, this is not a routine finding, and when both synovial structures are effusive, this finding will generally not apply.

In a study comparing ultrasound to low-field MRI for the equine digit, Evrard et al. [29] reported increased identification of suprasesamoidean DDFT lesions on ultrasound compared to standing, low-field (0.27T) MRI. It is difficult to know whether this finding was similar to ours in the over-diagnosis of suprasesamoidean tendon injuries with ultrasound or a failure of detection on MRI. Low-field MRI has greater limitations in resolution and slice thickness artifact compared to high-field MRI, resulting in a higher likelihood of milder lesions not being detected.

Distribution of lesions to the DDFT was similar to other studies [22], with enlargement and dorsal surface fraying or tearing occurring more commonly at the level of the proximal recess of the navicular bursa in 54 digits, compared to distally in 37 digits or proximally in the pastern in 14 digits. Horses with injury to the DDFT concurrently with additional structures within the digit have been shown to have a worse prognosis [15,21]. While the purpose of this study was not to report on outcome or prognosis, this has clinical implications for additional findings that may be missed with performing ultrasound alone. Therefore, if MRI is not feasible and only ultrasound is performed, it is important for both the clinician and client to understand the limitations in the diagnostic capability of this modality.

The comparison of DDFT findings in this study was confined to lesions within the hoof capsule at the level of P2 and did not include the images acquired in the pastern at the level of P1. Other work [30] has shown that lesions of the DDFT in the pastern are associated with lesions more distally in the tendon. Therefore, if a tear is identified more proximally in the pastern, this is a useful way to help increase confidence in the presence of a lesion in the suprasesamoiden region on the ultrasound exam. Additionally, that study showed that all horses with distal DDFT tears also had damage to the DDFT proximal to the navicular bone. In our study, there were seven horses that did not have suprasesamoidean DDFT abnormalities identified on ultrasound that had distal DDFT tears. This is concerning in that it indicates that not identifying a lesion of the DDFT proximal to the navicular bone also increases the chance of not recognizing the additional risk of a more distal DDFT tear. At the time of this study, our institution did not routinely include the mid-proximal pastern in a “foot” MRI. Now, with a more expanded recognition of the way in which DDFT injuries extend proximally, as well as the myriad other abnormalities that can be found in this area in horses that respond to a palmar digital nerve block, we have expanded our “foot study” MRIs to include images to the proximal aspect of the pastern/distal metacarpophalangeal joint.

The low numbers of distal interphalangeal joint collateral ligament injuries limited statistical analysis but did show a tendency toward overdiagnosis of damage with ultrasound. The fibers of the collateral ligaments of the distal interphalangeal joint have differing fiber orientations, which results in a central area of hypoechogenicity at a steeper beam angle [31]. The authors are aware of the fiber orientation and this artifact, and it was not misinterpreted. However, adjacent tissue can lead to indistinct surfaces of the ligament, leading to the risk of overinterpretation of enlargement. A learning effect did take place, however, with more of the over-diagnosed abnormalities occurring in the first half of the study.

A learning effect was also found in the ultrasonographic diagnosis of navicular bursal proliferation. In the first half of the study, the accuracy for navicular bursal proliferation was 49% and in the second half of the study was 74%. This analysis was performed retrospectively out of the interest of the investigators to see if they had improved in their accuracy over time. This improvement highlights the importance of experience, repetition, and comparison of modalities. Although both sonographers were experienced in scanning the distal limb prior to the exam, the repeated standardized studies with timely direct comparison to high-field MRI allowed the researchers to learn from their errors and improve interpretation. An interesting future direction of study would be to allow emerging and experienced sonographers to perform a similar study with temporally close direct comparison and assess how this changes the accuracy of ultrasound. The results of this study have further emphasized the ongoing value of comparing ultrasound findings to advanced imaging.

Although high-field MRI in many ways is superior to ultrasound for lesion detection in the hoof capsule, ultrasound does provide several advantages for lesion characterization. Due to the difference in tissue contrast characteristics between the two modalities, mineral within the tendon is more readily diagnosed with ultrasound than MRI [29]. Mineral; particularly, dystrophic or corticated is usually hypointense on MRI, as is tendon tissue. Thus, it can be quite difficult to distinguish minerals embedded within the tendon. In contrast, the mineral is quite hyperechoic on ultrasound, and depending on size, will usually cause distal acoustic shadowing. This allows it to be readily distinguished from the tendon, particularly when off-angle imaging causes the tendon to appear hypoechoic. Ultrasound also allows for dynamic evaluation of soft tissues in a way that is more difficult with MRI. This can be as simple as changing how the horse is standing to see how fluid moves or dynamically moving the limb to evaluate how this affects the appearance of soft tissues. Shifting from weighted to unweighted exams can help to better assess the extent and configuration of some types of lesions. Another benefit of ultrasound, while not utilized in this study, is that ultrasound allows the use of Doppler interrogation to help assess the vascularity of tendon lesions. The exact significance of increased Doppler signal still requires additional research, but it can be useful to help identify or confirm lesions and assess for active increased vascularity [32].

The study had several limitations. Transcuneal ultrasound was not performed, which could increase the detection of distal lesions of the DDFT; however, this is not clinically practical in an arid climate. Almost all of the horses in the study also had radiographs of the affected distal limbs. These were not included in the analysis as the study was primarily aimed at comparing soft tissue abnormalities, and because many of the radiographs were from referral facilities, they are thus not standardized. However, in a practical clinical setting, most patients receive radiographic evaluation when pain is localized to the distal limb. Assumedly, at least a portion of the navicular bone abnormalities found on MRI in this study would be identified radiographically. Therefore, there is a synergistic diagnostic effect of combining radiographs and ultrasound that was not addressed in this study. Selection bias can also be considered a limitation, as all horses were imaged due to lameness that was localized to the distal limb region. An additional limitation is using MRI as the gold standard. No histology was performed, and only a limited number of horses had bursoscopy performed. However, several studies have correlated changes to the DDFT and navicular bone on MRI to pathology on histology [13,16,18]. Further studies with larger sample sizes may also allow for better exploration of the distal interphalangeal joint collateral ligaments.

## 5. Conclusions

Many MRI studies have included horses in which a definitive diagnosis was not made with ultrasound. This study differs in that it is a direct, prospective comparison of ultrasound to MRI. The use of ultrasound for structures within the digit has been said to be limited [8], restricted to the evaluation of a change in shape or mineralization in the deep digital flexor. Hoof conformation and other patient factors can also affect image quality and diagnostic capability. However, in the authors’ opinion, ultrasound maintains value for a role in evaluating the equine digit. As long as the limitations are recognized, ultrasound can be useful as a non-invasive diagnostic tool for screening and initial diagnosis and potentially for monitoring recovery and response to treatment for certain lesions.

## Figures and Tables

**Figure 1 animals-13-02328-f001:**
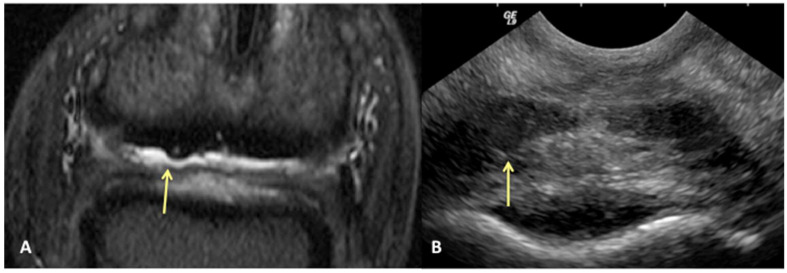
Transverse STIR MR image (**A**) and corresponding transverse US image (**B**), obtained at the level of the proximal recess of the navicular bursa. Palmar is on the top of the image, dorsal is on the bottom of the image, lateral is on the left. There is large area of bulging of the dorsal aspect of the lateral lobe of the deep digital flexor tendon (yellow arrows). All MR images are flipped to have the anatomy in the MR and ultrasound images in the same orientation.

**Figure 2 animals-13-02328-f002:**
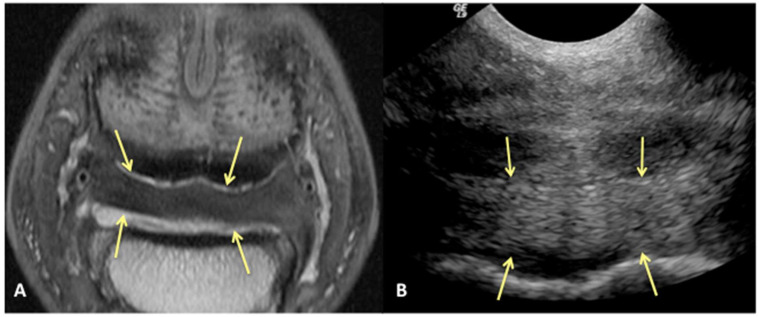
Transverse proton dense MR image (**A**) and corresponding transverse ultrasound image (**B**) of the collateral sesamoidean ligament. Palmar is on the top of the image, dorsal is on the bottom of the image, lateral is on the left. There is diffuse (mild) enlargement of the ligament (yellow arrows), positively identified with MRI and ultrasound.

**Figure 3 animals-13-02328-f003:**
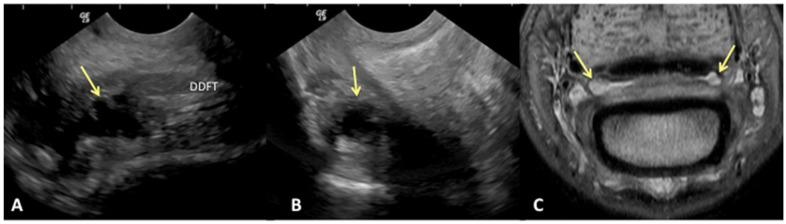
Transverse (**A**) and sagittal (**B**) ultrasound images of the lateral aspect of the proximal recess of the navicular bursa and corresponding transverse proton dense MR image (**C**). There is moderate cffusion and proliferation within the navicular bursa, identified on ultrasound and MRl (yellow arrows). (**A**) Lateral is on the left, palmar is on the top and dorsal on the bottom of the image. (**B**) Proximal is on the top of the image, distal is on the bottom. The palmar cortex of the middle phalanx is on the left of the image. (**C**). Palmar is on the top of the image, dorsal is on the bottom of the image.

**Figure 4 animals-13-02328-f004:**
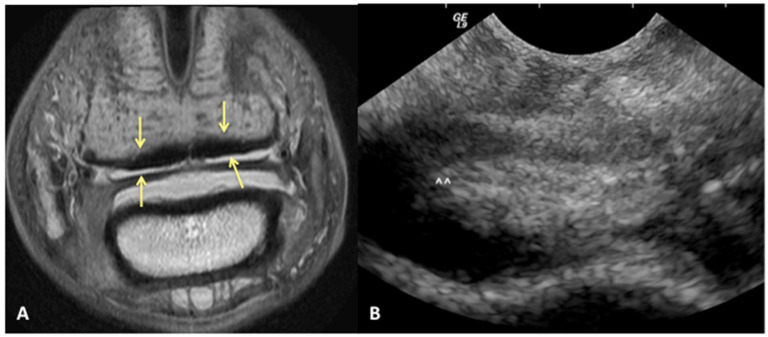
Transverse proton dense MR image (**A**) and corresponding ultrasound image (**B**) at the level of the collateral sesamoidean ligament and proximal recess of the navicular bone. Dorsal bulging and tearing was identified on ultrasound in the lateral lobe, white arrow heads; however, the DDFT is normal on MRI (yellow arrows). Palmar is on the top of the image, dorsal is on the bottom of the image, lateral is on the left.

**Table 1 animals-13-02328-t001:** Grading System for Evaluation of the Digit Using Ultrasound and MRI.

Structure	Ultrasound	MRI
Collateral Sesamoidean Ligament	Enlargement: present or absent Displacement: none, dorsal or palmar	Size: Normal, enlargement or thickening, Displacement- none, dorsal or palmar
Deep Digital Flexor Tendon	Symmetry of medial lateral lobes: asymmetric or symmetric Dorsal surface bulging or change in fiber pattern: present or absent Alterations in echogenicity (hyper or hypoechoic): present or absent	Asymmetry of the medial and lateral lobes- present or absent, Enlargement: 0–3 Dorsal surface fraying: 0–3 Dorsal surface tearing: 0–3 Core lesion at the level of the proximal recess of the navicular bursa: 0–3 Sagittal split at the level of the proximal recess of the navicular bursa: 0–3 Tearing proximal to the proximal recess of the navicular bone: 0–3 Tearing distal to the navicular bone: 0–3 Enthesopathy at the insertion on the distal phalanx: 0–3 Mineralization within the DDFT: present or absent
Navicular Bursa	Effusion: 0–3 Proliferation: 0–3	Effusion: 0–3 Proliferation and capsule thickening: 0–3
Navicular Bone	N/A	Signal changes: 0–3 Degenerative navicular bone changes (including flexor cortex erosions, sclerosis, increased size or number of the synovial invaginations): 0–3
Impar Ligament	N/A	Enlargement: 0–3 Enthesopathy or resorption on the distal phalanx at the site of the impar ligament insertion: 0–3

0: Normal, None, 1: Mild, 2: Moderate, 3: Severe.

**Table 2 animals-13-02328-t002:** Sensitivity and Specificity of Ultrasound versus MRI. (All comparisons are made at the level of the proximal recess of the navicular bursa.). MRI is considered the gold standard to which ultrasound is compared. Sensitivity is a measure of those with an MRI abnormality for which the ultrasound also detects the abnormality; specificity is a measure of those without an MRI abnormality of the given structure for which the ultrasound findings are also normal. Accuracy, or correct classification rate, is the proportion of observations for which the ultrasound diagnostic imaging findings and MRI findings agree, whether negative or positive.

Structure: US vs. MRI	Sensitivity	Specificity	Accuracy
Asymmetry of the DDFT	77.27 17/22	70.83 34/48	72.86
Dorsal Damage/Enlargement (US) vs. Dorsal Tearing (MRI)	85.19 23/27	60.47 26/43	70.00
Dorsal Damage/Enlargement (US) vs. Fraying MRI	62.96 34/54	62.50 10/16	62.86
Navicular bursa Effusion (US) vs. Effusion (MR)	64.71 22/34	69.44 25/36	67.14
Navicular Bursa Proliferation (US) vs. Proliferation (MRI)	47.37 18/38	78.13 25/32	61.43
CSL Enlargement (US vs. MRI)	42.42 14/33	78.38 29/37	61.43
CSL Displacement (US vs. MRI)	35.29 6/17	81.13 43/53	70

## Data Availability

Not applicable.

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
