# Peer review of "Comparison of Ultrasound and Magnetic Resonance Imaging for Identifying Soft Tissue Abnormalities in the Palmar Aspect of the Equine Digit"

_animals, 2023, doi:10.3390/ani13142328_

Round 1
Reviewer 1 Report
The manuscript entitled "Comparison of ultrasound and magnetic resonance imaging for identifying soft tissue abnormalities in the palmar aspect of the equine foot" presents a comparison of two diagnostic methods used in equine orthopedics. My overall substantive assessment of the article is high. It is a study extremely useful in clinical practice and at the same time the results are supported by a solid scientific workshop. Including detailed comments in the review would be quite difficult, because the copy submitted for review does not contain line numbers or even page numbers. Fortunately, the manuscript does not contain significant errors requiring correction. Below are my suggestions that should increase the value of the work: Statistical methods should be described in more detail in the chapter "Matherials and Methods". The terms "sensitivity, specifity, accuracy" should be clarified. How was the p-value calculated? My minor remarks concern the anatomical terminology. The terms "palmar" and "foot", present in the title and many times in the text, are basically mutually exclusive. " is used for the distal part of the thoracic limb, while "foot" (Lat. pes) means the distal part of the pelvic limb. I am aware that the terminology used by the authors is common in the clinical literature, but is incorrect from the point of view of comparative anatomy. I suggest replacing the word "foot" with the term "distal limb" or even "digit" (the considerations do not go beyond this part). It is also worth explicitly declaring that only the thoracic limbs were examined. Another minor reservation concerns the lack of explanation of the abbreviation "NB". "Dorsal margin" would prefer to replace it with "dorsal surface" due to the shape of the tendon of insertion. Please explain what is "medial lobe" vs. "lateral lobe". I don't understand the term "proximal recess of the navicular bone" used in the discussion. Also, I don't know how pathology of the navicular bone can include "increased fluid signal". Finally, I would like to note that the minimum suggested word count for the paper type "article" is 4,000, while the reviewed manuscript contains only about 2,700 words.
Author Response
Thank you for taking the time to review the manuscript and make suggestions to improve it.
Statistical methods should be described in more detail in the chapter "Matherials and Methods". The terms "sensitivity, specifity, accuracy" should be clarified.
This description has been made more robust
How was the p-value calculated?
Thank you for pointing this out. The reported p-value was to compare to the null hypothesis of 50% and thus was of low relevance. We have removed it to make the table more clear.
My minor remarks concern the anatomical terminology. The terms "palmar" and "foot", present in the title and many times in the text, are basically mutually exclusive. " is used for the distal part of the thoracic limb, while "foot" (Lat. pes) means the distal part of the pelvic limb. I am aware that the terminology used by the authors is common in the clinical literature, but is incorrect from the point of view of comparative anatomy. I suggest replacing the word "foot" with the term "distal limb" or even "digit" (the considerations do not go beyond this part). It is also worth explicitly declaring that only the thoracic limbs were examined.
The terminology has been changed as requested to digit and distal limb throughout.
Another minor reservation concerns the lack of explanation of the abbreviation "NB".
Spelled out to "navicular bursa"
"Dorsal margin" would prefer to replace it with "dorsal surface" due to the shape of the tendon of insertion.
Changed throughout
Please explain what is "medial lobe" vs. "lateral lobe".
Explanation of the symmetric halves of the ddft has been added.
I don't understand the term "proximal recess of the navicular bone" used in the discussion.
Thank you for seeing this typo. Bone has been replaced with "bursa"
Also, I don't know how pathology of the navicular bone can include "increased fluid signal".
This has been clarified as increased signal on STIR sequences, also known as bone marrow lesions or bone edema-like syndrome
Finally, I would like to note that the minimum suggested word count for the paper type "article" is 4,000, while the reviewed manuscript contains only about 2,700 words.
The text is now approximately 4200 words. Thank you again.
Reviewer 2 Report
Dear Authors,
This is a well constricted paper which is useful in that it corroborates findings from retrospective comparative imaging studies in terns of the accuracy of ultrasound compared with HF MRI as a gold standard, which has been well validated in previous studies.
It would have been nice to have a larger sample size, and to have been able to include findings for the collateral ligaments of the DIP joint for example, but the information provided is useful clinically as a reminder of the limitations of ultrasound in the foot region, and of its role in monitoring healing for example.
Best wishes to you

Author Response
Thank you very much for you kind comments and feedback. We have added small sample size as a limitation and further assessment of the distal interphalangeal joint collateral ligaments as a further directional aim, as we are in complete agreement with your statement.